# The bi-factor structure of the 17-item Hamilton Depression Rating Scale in persistent major depression; dimensional measurement of outcome

Neil Nixon[1,2], Boliang Guo[1,3], Anne Garland[2], Catherine Kaylor-Hughes[1,3], Elena Nixon[1,3], Richard Morriss[1,3] *

**1** Division of Psychiatry and Applied Psychology, School of Medicine, University of Nottingham, Nottingham, United Kingdom, **2** Adult Mental Health Directorate, Nottinghamshire Healthcare Trust, Nottingham, United Kingdom, **3** ARC EM, School of Medicine, University of Nottingham, Nottingham, United Kingdom

* mczrkm@exmail.nottingham.ac.uk

**Data Availability Statement:** All relevant data are within the paper and its Supporting Information files.

## Abstract

### Background

The 17-item Hamilton Depression Rating Scale (HDRS$_{17}$) is used world-wide as an observer-rated measure of depression in randomised controlled trials (RCTs) despite continued uncertainty regarding its factor structure. This study investigated the dimensionality of HDRS$_{17}$ for patients undergoing treatment in UK mental health settings with moderate to severe persistent major depressive disorder (PMDD).

### Methods

Exploratory Structural Equational Modelling (ESEM) was performed to examine the HDRS$_{17}$ factor structure for adult PMDD patients with HDRS$_{17}$ score $\geq$16. Participants (n = 187) were drawn from a multicentre RCT conducted in UK community mental health settings evaluating the outcomes of a depression service comprising CBT and psychopharmacology within a collaborative care model, against treatment as usual (TAU). The construct stability across a 12-month follow-up was examined through a measurement equivalence/invariance (ME/I) procedure via ESEM.

### Results

ESEM showed HDRS$_{17}$ had a bi-factor structure for PMDD patients (baseline mean (sd) HDRS$_{17}$ 22.6 (5.2); 87% PMDD >1 year) with an overall depression factor and two group factors: vegetative-worry and retardation-agitation, further complicated by negative item loading. This bi-factor structure was stable over 12 months follow up. Analysis of the HDRS$_{6}$ showed it had a unidimensional structure, with positive item loading also stable over 12 months.

### Conclusions

In this cohort of moderate-severe PMDD the HDRS$_{17}$ had a bi-factor structure stable across 12 months with negative item loading on domain specific factors, indicating that it may be

**Funding:** The data collected for this report was funded by 2 centre grants from the National Institute for Health Research (NIHR) Collaboration for Leadership in Applied Research and Care Nottinghamshire, Derbyshire and Lincolnshire (awarded to Manning N, Morriss R, Cooke M, Currie G, Hollis C, Kai J, Schneider J, Walker M), and the NIHR Collaboration for Leadership in Applied Research and Care East Midlands (awarded to Khunti K, Morriss R, Singh S, Gladman J, Waring J. Since these are centre grants they do not have grant numbers but are identified by the name of the grant. The grants were awarded by the National Institute for Health Research whose URL is https://www.nihr.ac.uk The funders had no role in study design, data collection and analysis, decision to publish, or preparation of the manuscript.

**Competing interests:** The authors have declared that no competing interests exist.

more appropriate to multidimensional assessment of settled clinical states, with shorter uni-dimensional subscales such as the HDRS$_6$ used as measures of change.

## Introduction

The 17-item Hamilton Depression Rating Scale (HDRS$_{17}$), which was developed in late 1950s has been the most frequently used observer-rated measure of depression for research including randomised controlled trials (RCTs) of treatments for depression [1–3]. The positive and negative features of the HDRS$_{17}$ have been comprehensively reviewed [4–6]. One of its most serious problems, poor inter-rater and test-retest reliability has been addressed with the development of the GRID- HDRS$_{17}$ version [5]. Overall despite its flaws, it continues to be recommended by licensing and treatment guideline bodies such as the Federal Drug Administration in the US [7] and the National Institute for Care Excellence [8] because of its longitudinal continuity for historical comparison in more than 1500 randomised controlled trials, widespread use for meta-analysis and the lack of a superior measure despite many attempts and considerable resources including the National Institute of Mental Health and the World Health Organisation [4–6].

However, concerns persist about the widespread use of the of the HDRS$_{17}$ as a unidimensional measure of depression severity, given indications that it has a more complex factor structure that is not fully captured by a single, total score [9–13]. Evidence supporting a multidimensional structure has been reviewed by Fried et al [9] and demonstrated across different methodologies, including hierarchical confirmatory factor analysis (CFA) showing a general 2nd order depression factor [13] and exploratory factor analysis in 'treatment naïve' mainly non-persistent depression [14, 15]. Fried et al [9], went further to show that this multifactorial structure became more pronounced as depression severity increased, indicating the potential importance of assessing HDRS$_{17}$ structure in clearly defined clinical groups identified by levels of persistence and severity.

In fact, since Hamilton's original factor analysis [2], relatively little work has been done on the HDRS$_{17}$ factor structure in patient groups with more severe, persistent major depressive disorder (PMDD) under treatment in mental health settings. Findings have instead emerged from a variety of other clinical settings [11] and populations, including people whose primary health problem was not depression [12]; and since the nature and complexity of depression has been shown to vary widely across these populations, including the degree of persistence, melancholia, anxiety and other associated co-morbidity [16–18], it follows from Fried et al [9] that these findings cannot be assumed to give a true impression of how the HDRS$_{17}$ functions within more severe PMDD. Additionally, methods of statistical analysis have changed over the 60 years since Hamilton's original work and earlier reports often lacked the more robust analytical approach now available for establishing factor structure through Exploratory Structural Equational Modelling (ESEM) [19].

Current modelling via ESEM also allows assessment of the related issue of measurement invariance, assessed through the consistency of construct measurement across time. Whilst Fried et al [9] showed this was generally poor for a range of depression measures including the HDRS$_{17}$, more recent work using ESEM has shown invariance over 12 months for a patient completed outcome measure in more severe PMDD (the 9-item Personal Health Questionnaire; PHQ-9) [15, 20]. However, there has been no equivalent assessment of clnician outcome measures, such as the HDRS$_{17}$ in this PMDD popultation.

Previous statistical approaches assessing the dimensionality of the $HDRS_{17}$ have included both exploratory factor analysis (EFA) and CFA [13]. However, recent literature has shown that both EFA and CFA have methodological limitations [21, 22]. In EFA modelling it is impossible to incorporate latent EFA factors into subsequent analyses and it is not easy to test measure invariance across groups and/or times [22]. In CFA modelling, each item is strictly allowed to load on one factor and all non-target loadings are constrained to zero. The latest factor analytical ESEM, integrates the best features of both EFA and CFA together by applying EFA rigorously to specify more appropriately the underlying factor structure together with the advanced statistical methods typically associated with CFAs [22]. ESEM allows cross item factor loadings which are coherent with the underlying theory and/or item contents that item(s) could cross load on different latent factors; ESEM could reduce the bias in parameter estimates due to zero loading restriction which generally results in inflated CFA factor correlation because items might not be perfect factor indicators with some degree of irrelevant association with other constructs [22–24].

Following on from these findings, in order to explore whether the $HDRS_{17}$ has a general depression factor and additional domain specific factors a bi-factor model exploring psychometric multidimensionality within ESEM, is now recommended, rather than the traditional second order factor analytical model [23, 25, 26]. Compared with hierarchical factor analysis model (Fig 1), bi-factor models have statistical advantages such as fitting data better and allowing external prediction by group factors with or without overall factors [27]. Conceptually, as group factors in a bi-factor model are not subsumed by the overall factor [28], they represent factors explaining items variances which were not accounted for by the overall factor [27]. Therefore the group specific factors have influence over and above the general factor that might help explain the clinical heterogeneity observed among individual patients with depression [29], providing valuable clarity for future research and practice.

There is therefore an opportunity and need to re-assess the factor structure and measurement invariance of the $HDRS_{17}$ in more severe PMDD, made more pressing by the fact that treatment guidelines, including those currently in preparation [30], continue to use the $HDRS_{17}$ as a single total score across a range of depression severity and persistence.

We address this issue here via ESEM bi-factor modelling in a well-defined patient population with moderate to severe PMDD, recruited from UK mental health care settings in a previously published RCT [18], assessing construct stability across 12-month follow-up using a measurement equivalence/invariance (ME/I) procedure. The chosen 12-month period is

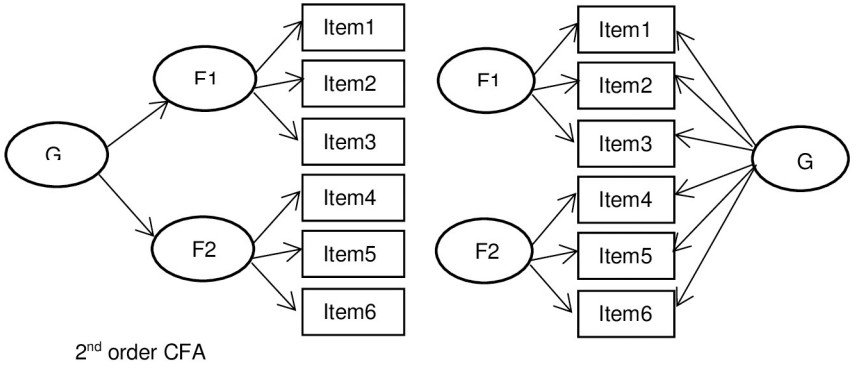

**Fig 1. Schematic example of 2ⁿᵈ order factor and bi-factor model: G = general factor, F = group factor.**

clinically relevant through the extended clinical treatment often necessary in patients with PMDD.

## Materials and methods

### Patients and instruments

Patients ($N$ = 187) were drawn from a multicentre pragmatic randomised controlled trial (RCT) evaluating outcomes of a Special Depression Service (SDS; specialist pharmacotherapy and psychotherapy within a collaborative care model) against treatment as usual (TAU) within UK mental health services [18]. At the time of recruitment participants were all adults receiving community treatment for persistent depression in one of three UK mental health centres (Nottingham, Derby and Cambridge). Ethics approval was obtained from the Trent Research Ethics Service in Derby, England. Approval number 09/H0405/42. Oral and written informed consent was obtained from each participant.

Participants were eligible for the study if they were: thought by the referrer to have primary unipolar depression; aged 18 years or over; able and willing to give oral and written informed consent to participate in the study; had been offered or received direct and continuous care from one or more health professionals in the preceding 6 months and currently be under the care of a secondary care mental health team; had a diagnosis of major depressive disorder with a current major depressive episode according to the structured clinical interview for DSM-IV (SCID) [31]; met five of nine NICE criteria for symptoms of moderate depression; had a score of ≥16 on the 17-item GRID version of the Hamilton Depression Rating Scale (HDRS17) [5]; and had a Global Assessment of Functioning (GAF) [32] score ≤ 60. Referrals were excluded if they: were in receipt of emergency care for suicide risk; were at risk of severe neglect, or posed a homicide risk, unless that risk was adequately contained in their current care setting; were not fluent English speakers; were pregnant; had unipolar depression secondary to a primary psychiatric or medical disorder, except when bipolar disorder was identified by the research team after referral with unipolar depression because an SDS would be expected to manage bipolar depression in clinical practice (n = 8, 4.3%).

The mean age of patients was 46.8 years (sd 11.4) and 61.1% (114 of total 187) were female. Following randomisation 93 (49.7%) patients were allocated to the SDS treatment arm and 94 (50.3%) to treatment as usual (TAU). In the treatment arm, participants received specialist pharmacological and cognitive behaviour therapy within a collaborative care model structured and planned over 12 months. TAU comprised multidisciplinary, community-based care delivered by general mental health services. The primary clinical outcome measure in this trial was the $HDRS_{17}$ assessed at baseline, 6 and 12 month follow up time points [33]. One hundred and sixty-three (87%) participants entering the RCT suffered depression for more than 1 year with the median (interquartile range) duration of the current episode of 6.5 (2.6–16.0) years. The mean (sd) severity of the $HDRS_{17}$ at baseline was 22.6 (5.2) years. Melancholia was present in 105 (56.1%) participants and 146 (78.1%) also had a comorbid anxiety disorder. The study design, data collection procedures, treatment offered and trial results can be found from the published protocol [33] and trial report [18].

The $HDRS_{17}$ evaluates depression severity through items on: 1) depressed mood, 2) guilt, 3) suicidal thought or action, 4) insomnia initial, 5) insomnia middle, 6) insomnia late, 7) work and interests (assessing pleasure and functioning), 8) motor retardation, 9) motor agitation, 10) psychic anxiety, 11) somatic anxiety, 12) appetite, 13) tiredness, 14) sexual interest, 15) hypochondriasis, 16) weight loss, 17) insight. Among these 17 items, 9 items are scored on a 5-point scale (0–4) and 8 items on 3-point scale (0–2) with higher scores indicating greater depressive severity for all items. In keeping with current practice, the total item

score was used to quantify the severity of depression and treatment effect estimates in the RCT [2].

## Statistics

We first examined the frequency of patients' response on each $HDRS_{17}$ item across three time points (baseline, 6 and 12 months). ESEM was then used to explore the factor structure of the $HDRS_{17}$ [22]. With reference to existing evidence on the factor structure of the $HDRS_{17}$, we tested separately one to five first order factors and also bi-factor models with two-three domain specific factors for data measured at each time point. Data measured at each time point were stored in wide format for ESEM modelling with alike items measured at adjacent time correlated to take into account the non-independence of data due to the nature of longitudinal design [34]. Ordinal item score was analysed with the WLSMV estimator using Delta parameterization; missing values were automatically accounted for using the full-information maximum likelihood approach built into Mplus [35, 36]. Measurement invariance across all follow-up time points for the best fitted factor structure was further tested using ESEM by comparing configural invariance model and scalar invariance (item factor loading and item threshold invariance) model fittings [9, 34, 37]. All ESEM models were performed using software Mplus 8 and in keeping with standard practice correlation between item residuals was set as 0 [37].

Several fitting indices along with chi-square ($\chi^2$) test were used to judge model fit as $\chi^2$ tests are sensitive to large sample sizes and non-normal data [38]. The criterion are both comparative fit index (CFI) and the non-normed fit index (NNFI) > 0.90, Root Mean Square Error of Approximation (RMSEA) < 0.08 [39]. The factor loading and item-factor mapping pattern were additionally examined by two senior psychiatrists (RM, NN) to make the factor structure clinically plausible and meaningful. Model comparisons were evaluated by reference to the $\chi^2$ change test using Mplus DIFFTEST function to conduct $\chi^2$ difference tests, as the WLSMV estimator was used to analyse ordinal items scores [37]. Since the $\chi^2$ change tests are influenced by sample size and data non-normality [34, 40, 41], the CFI change is independent of both model complexity and sample size and it is not correlated with the overall fit measurements. A reduction of 0.01 or more in CFI suggests the null hypothesis of no difference should be rejected [41]. We therefore mainly judged model improvement on the CFI change [34, 41] A number of specific modelling details are presented alongside the results.

## Results

### Frequency of item response

The frequency of each item by arm across measurement time are presented as an appendix. There is an extreme response pattern for the item "insight loss", for which all but one response was recorded as 0 across measurement time. This extreme response on item "insight loss" would result in it being excluded from all ESEM modelling due to 0 variability. Hence all ESEM models in this study were performed using 16 items.

### $HDRS_{17}$ factor structure

Model fitting indices of structure included one to five first order factors and bi-factor models with two or three domain specific factors for measures at each time (Table 1). Although the model fitting increased with an increased number of latent factors, the items-factors association mapping showed that the bi-factor model with two domain specific factors (bi-2factor) had the most meaningful factor structure in term of model fitting and item-factor mapping pattern. A similar pattern was shown when all models in Table 1 were rerun with alike item

**Table 1. Modelling fitting indices for model with different 1st order and bi-factor structures.**

| Model | $\chi^2$(df),p = | RMSEA | CFI | NNFI | ΔCFI | Δ$\chi^2$(df),p = |
|---|---|---|---|---|---|---|
| 1-factor | 1375.249(1045), 0.000 | .041 | .812 | .797 | | |
| 2-factor | 1213.822(991), 0.000 | .035 | .873 | .856 | .61 | 159.470(54), 0.000 |
| 3-factor | 1079.051(934), 0.001 | .029 | .917 | .900 | .44 | 142.193(57), 0.000 |
| Bi-2factor# | 1079.051(934), 0.001 | .029 | .917 | .900 | | 142.193(57), 0.000 |
| 4-factor | 949.400(874), 0.038 | .021 | .957 | .945 | .40 | 139.142(60), 0.000 |
| Bi-3factor# | 949.400(874), 0.038 | .021 | .957 | .945 | | 139.142(60), 0.000* |
| 5-factor | 839.582(811), 0.236 | .014 | .984 | .977 | .27 | 124.888(63), 0.000 |

#Bi-2(3) factor model has same fitting indices as 3(4) factor model.

*Comparing with bi-2factor model.

loading set to be equal across measurement time (Table 2). The item-factor association mapping also showed that a bi-factor model with two domain specific factors (Table 4) had the most meaningful factor structure (Table 3). By examining the factor loading pattern shown in Table 3, it was suggested HDRS$_{17}$ measured a general depression factor for patients with moderate-severe PMDD, which comprised all items except "motor retardation" together with a vegetative-worry factor comprising positively loading items "insomnia" (early, middle and late), 'weight loss", "appetite loss" and negative loading items "psychic anxiety" and "hypochondriasis"; and a retardation-agitation factor comprising positive loading items "motor retardation", "depressed mood", diminished pleasure ("work and interests"), "suicidal thoughts" and negative loading for "agitation". Item factor loadings for all models shown in Table 2 are presented as supplementary material (appendix).

## Stability of factor structure across measure time

The fitting indices of ME/I test models for configural and scalar invariance across measurement time are presented for comparison in Table 4, indicating that the scalar invariant model should be retained as the CFI drop is 0.001 with $\chi^2$ increase at 147.674 (df = 119), p = 0.038. These results evidence that the bi-2factor structure is stable through follow up from baseline to 6 and 12 months.

In view of this stable but complex bi-2factor structure, including negative item loadings on both domain specific factors, we conducted a further post-hoc analysis of the most commonly used HDRS subscale, the HDRS$_6$ in the same cohort to investigate its potential as an alternative

**Table 2. Modelling fitting indices for various models with equal loading across measurement time.**

| Model | $\chi^2$(df),p = | RMSEA | CFI | NNFI | ΔCFI | Δ$\chi^2$(df),p = |
|---|---|---|---|---|---|---|
| 1-factor | 1372.823(1075), 0.000 | .038 | .831 | .822 | | |
| 2-factor | 1247.154(1047), 0.000 | .032 | .886 | .877 | | 111.539(28), 0.000 |
| 3-factor | 1143.753(1012), 0.002 | .026 | .925 | .916 | .49 | 106.877(35), 0.000 |
| Bi-2factor# | 1143.753(1012), 0.002 | .026 | .925 | .916 | .49 | 106.877(35), 0.000 |
| 4-factor | 1058.244(970), 0.025 | .022 | .950 | .942 | .25 | 90.859(42), 0.000 |
| Bi-3factor# | 1058.244(970), 0.025 | .022 | .950 | .942 | .25 | 90.859(42), 0.000* |
| 5-factor | 969.202(921), 0.131 | .017 | .973 | .966 | .17 | 103.371(49), 0.000 |

#Bi-2(3) factor model has same fitting indices as 3(4) factor model.

*Comparing with bi-2factor model.

**Table 3. Factor loading of best fitted model.**

| Item | Vegetative Worry | General depression | Retardation Agitation |
|---|---|---|---|
| depressed mood | -0.072 | **0.440** | **0.342** |
| guilt feeling | -0.069 | **0.391** | 0.130 |
| suicidal thoughts | -0.006 | **0.381** | **0.260** |
| insomnia initial | **0.478** | **0.181** | -0.018 |
| insomnia middle | **0.636** | **0.239** | 0.072 |
| insomnia delayed | **0.465** | **0.192** | 0.043 |
| work & interests | 0.111 | **0.380** | **0.322** |
| motor retardation | 0.097 | 0.054 | **0.601** |
| Agitation | -0.036 | **0.336** | **-0.366** |
| psychic anxiety | **-0.302** | **0.546** | -0.003 |
| somatic anxiety | -0.098 | **0.486** | 0.008 |
| appetite decrease | **0.281** | **0.399** | -0.070 |
| Tiredness | 0.07 | **0.519** | 0.069 |
| sexual interest | -0.008 | **0.268** | 0.122 |
| Hypochondriasis | **-0.259** | **0.328** | -0.106 |
| weight loss | **0.386** | **0.352** | -0.351 |

# estimate in bold statistically significant at p<0.05.

change measure to the full HDRS$_{17}$ in moderate-severe PMDD [42]. The HDRS$_6$ comprises 6 items: *depressed mood*, *work and interests (pleasure)*, *general somatic (tiredness)*, *psychic anxiety*, *guilt feelings and psychomotor retardation*; and since it was not plausible to perform an exploratory analysis testing a model with 1 to 3 factors on a 6-item scale, we instead used a one factor model to test its unidimensional factor structure. Results given in Tables 5 and 6 show that all 6 items of the HDRS$_6$ subscale loaded positively and significantly, with time invariance; supporting this as a stable, unidimensional outcome measure in moderate-severe PMDD, in contrast to the 17-item scale.

## Discussion

In light of findings that the HDRS$_{17}$ is not a unidimensional measure of depression [9, 14, 43, 44], that the factor structure may differ between clinical populations [9] and may not be stable over time, we aimed to assess the HDRS$_{17}$ in a well-defined group of patients with moderate to severe PMDD, using contemporary ESEM modelling. Consistent with much of this earlier work, our results in moderate-severe PMDD showed that the HDRS$_{17}$ had a bi-factor, rather than unidimensional structure. We additionally showed that this structure was time-invariant through the full 12-month period of study. The bi-factor structure comprised a general depression factor and two domain specific factors, which we refer to as 'vegetative-worry' and 'retardation-agitation'. The bi-factor structure was further complicated by the two domain specific factors including both positively and negatively loading items, problematising use of the HDRS$_{17}$ as an outcome measure in moderate to severe PMDD—even allowing for multiple

**Table 4. Fitting indices of ME/I across measurement time.**

| Model | χ²(df),p = | RMSEA | CFI | NNFI | ΔCFI | Δχ²(df),p = |
|---|---|---|---|---|---|---|
| Configural | 1079.051(934), 0.001 | .029 | .917 | .900 | | |
| Scalar | 1201.002(1053),0.001 | .027 | .916 | .910 | .001 | 147.674 (119), p = 0.038 |

**Table 5. Factor loading for HDRS$_6$ subscale.**

| Item | HDRS$_6$ |
|---|---|
| Depressed Mood | .544* |
| Work and Interests | .526* |
| General Somatic (Tiredness) | .474* |
| Psychic Anxiety | .417* |
| Guilt Feelings | .396* |
| Psychomotor retardation | .407* |

* all loading estimates statistically significant at p<0.01.

domain scoring within a bi-factor structure, we are left with the problem of incorporating domain factor items with opposite directionality. This problem was previously encountered within development of the 6-item subscale (HDRS$_6$) where agitation was excluded due to reciprocal interaction with the other items [45]; and opposite directionality cannot be surprising when applying the GRID-HDRS$_{17}$ to severe PMDD, when severe retardation is described by as 'all movements very slowed' and severe agitation as 'cannot sit still. . .pacing' [5].

Given these findings on the complex multidimensional structure of the HDRS$_{17}$ in moderate-severe PMDD and the associated question of its legitimacy as an outcome measure for this patient group, we ran a further post-hoc analysis of the most commonly used 6-item subscale to test its dimensionality and potential as an alternative measure of change to the 17-item scale [42]. The HDRS$_6$ subscale was derived through item analysis of the HDRS$_{17}$ against global assessment of depression by experienced psychiatrists and it has already demonstrated a unidimensional structure in some clinical populations [43, 45]. Our results confirm this unidimensionality in moderate-severe PMDD, additionally showing time-invariance over 12 months; supporting use of the HDRS$_6$ as an appropriate outcome measure in this group. In contrast our findings on the HDRS$_{17}$ do not support its use in this way.

What then for the 17-item scale? Firstly, it seems likely that this was initially conceived as a state measure, rather than a measure of change [2]. It's more complex structure, including concepts now understood as near polar opposites (e.g. agitation and retardation as operationalised in the GRID-HDRS$_{17}$) may still be more relevant to the assessment of settled clinical states, where the domain factors we have identified may further clarify depression type, acting as predictor variables to assist development of treatment strategies [45]. A patient loading high on worry (psychic anxiety, hypochondriasis) rather than vegetative disturbance (sleep, appetite, weight), may for example benefit from more targeted initial clinical interventions reflecting this delineated state rather than non-specific depression treatments [46]. The HDRS$_{17}$ might then be repeated later on for this individual, not as a measure of change, but to re-conceptualise a later settled state (such as a limited but stable treatment response) in order to develop next-step treatment strategies–in this model outcome change would be assessed through more parsimonious, evidence-based item-sets, such as the HDRS$_6$.

**Table 6. Fitting indices of ME/I across measurement time, HDRS$_6$ subscale.**

| Model | $\chi^2$(df),p = | RMSEA | CFI | NNFI | ΔCFI | $\Delta\chi^2$(df),p = |
|---|---|---|---|---|---|---|
| Configural | 193.266(120),0.000 | .057 | .934 | .913 | | |
| Scalar a | 459.218(165),0.000 | .098 | .736 | .755 | | 259.790(45), p = 0.000 |
| Scalar b* | 229.625(146),0.000 | .055 | .925 | .921 | -.009 | 44.883(26), p = 0.012 |

* scalar b model freed 24 of 55 (43%) threshold parameters estimates.

This approach seems in keeping with the initial history of the Hamilton scale. An awareness of the multidimensionality of the $HDRS_{17}$ dates back 60 years to Hamilton's original work, also based in observations on patients suffering severe depression within mental health treatment; identifying four hierarchical factors ("general", "endogenous", "anxious" and "insomnia") that show parallels with the bi-factor model derived here; including a main "general depression" factor, a retarded-depressed factor, a broadly vegetative factor and a separate factor including psychic anxiety [2]. Subsequent use of the $HDRS_{17}$ to report a single, total item score risks missing the potential richness and purpose of this scale; confirmed again by the ESEM structure presented here. Similarly, use of the $HDRS_{17}$ to measure change seems both unintended and unsupported by the growing evidence base.

The strengths of our study include a well characterised sample; the systematic application of a standardised interview version of the $HDRS_{17}$; the multicentre design; and assessment over three-time intervals across 12 months with adequate retention. The systematic application of both psychiatric and psychological treatment over this time period in one group versus usual care provided both a test of the robustness of the factor structure of the $HDRS_{17}$ and data from a broad group within UK mental health service care. Analysis included the first use of the most advanced ESEM modelling which allows cross factor loading and bi-factor modelling to simultaneously explore the overall latent factor and specific sub-factors for PMDD patients $HDRS_{17}$ measures, incorporating the ME/I test of invariance [22, 23, 40].

Our findings on the $HDRS_{17}$ and $HDRS_{6}$ are however limited to a single UK cohort of patients with moderate-severe PMDD. Given previous findings that factor structure may change with clinical characteristics of depression, such as severity [9], our findings do not presume that the same structure holds for other populations with less persistent, complex or severe depression. This caution fits with recognised features of PMDD, such as rumination/worry [47] and high comorbidity (e.g. 78.1% of the current cohort had a separate anxiety disorder), which may not be present in less severe, less persistent depression. Equally, psychomotor disturbance (through agitation or retardation) identified within our PMDD cohort may be much less prevalent in patients recruited from primary care or other general medical settings [1, 17]. It seems quite plausible in this regard that a different factor structure may emerge in these different clinical groups and whilst our preliminary findings in PMDD remain important, they cannot be assumed to generalise. Rather, important differences between clinical groups may be reflected in real changes to the underlying factor structure of the measurement tool, accounting for some observed differences between this and earlier studies conducted in predominately non-persistent depression [14]. Other important limitations include: the lack of a specific power calculation for the purposes of the current analysis [18, 33], though its size was sufficient to perform factor analysis modelling based in previous work on the methodology used here [48]; and the 40 per cent attrition over 12 months follow up, though again this left a sufficient sample for invariance analysis.

Data from the current study could be meta-analysed in future with other studies with similar designs and analysis methods to provide more robust results on the $HDRS_{17}$ factor structure in patients with moderate-severe PMDD.

## Conclusions

These preliminary findings in patients with moderate-severe PMDD indicate the $HDRS_{17}$ has a bi-factorial structure characterised by a general depression factor with two additional factors, 'vegetative-worry' and 'retardation-agitation'. This conceptual structure was found to be relatively stable across a 12-month follow up period but negative item loading on the $HDRS_{17}$ domain specific factors does not support its use as an outcome measure in this clinical

population. Instead, the $HDRS_{17}$ may be more appropriate in the multidimensional assessment of settled clinical states, helping to guide targeted interventions; with shorter unidimensional subscales such as the $HDRS_6$ used as measures of change.

## Supporting information

**S1 Data.**
(CSV)

**S2 Data.**
(DOCX)

## Acknowledgments

We wish to acknowledge the contribution of all the staff who obtained the data and participants in the original randomised controlled trial from which this analysis was conducted.

## Author Contributions

**Conceptualization:** Boliang Guo, Richard Morriss.

**Data curation:** Neil Nixon, Anne Garland, Catherine Kaylor-Hughes.

**Formal analysis:** Boliang Guo.

**Funding acquisition:** Anne Garland, Richard Morriss.

**Investigation:** Elena Nixon.

**Methodology:** Boliang Guo.

**Project administration:** Catherine Kaylor-Hughes, Elena Nixon.

**Resources:** Neil Nixon, Richard Morriss.

**Supervision:** Richard Morriss.

**Validation:** Neil Nixon.

**Writing – original draft:** Boliang Guo.

**Writing – review & editing:** Neil Nixon, Anne Garland, Catherine Kaylor-Hughes, Elena Nixon, Richard Morriss.

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
