## [Decision Letter · Decision Letter 0]

5 May 2020

PONE-D-20-01015

The bi-factor structure of the 17-item Hamilton Depression Rating Scale with persistent major depression disorder in specialist mental health services.

PLOS ONE

Dear Prof Morriss,

Thank you for submitting your manuscript to PLOS ONE. After careful consideration, we feel that it has merit but does not fully meet PLOS ONE’s publication criteria as it currently stands. Therefore, we invite you to submit a revised version of the manuscript that addresses the points raised during the review process.

We would appreciate receiving your revised manuscript by Jun 19 2020 11:59PM. To enhance the reproducibility of your results, we recommend that if applicable you deposit your laboratory protocols in protocols.io, where a protocol can be assigned its own identifier (DOI) such that it can be cited independently in the future. For instructions see: http://journals.plos.org/plosone/s/submission-guidelines#loc-laboratory-protocols

We look forward to receiving your revised manuscript.

Kind regards,

Thach Duc Tran, M.Sc., Ph.D.

Academic Editor

PLOS ONE

Journal Requirements:

Reviewers' comments:

Reviewer's Responses to Questions

**Comments to the Author**

1. Is the manuscript technically sound, and do the data support the conclusions?

Reviewer #1: No

Reviewer #2: Yes

2. Has the statistical analysis been performed appropriately and rigorously? 

Reviewer #1: Yes

Reviewer #2: Yes

3. Have the authors made all data underlying the findings in their manuscript fully available?

Reviewer #1: No

Reviewer #2: Yes

4. Is the manuscript presented in an intelligible fashion and written in standard English?

Reviewer #1: Yes

Reviewer #2: Yes

5. Review Comments to the Author

Reviewer #1: I welcome the initiative of the authors to explore psychometric properties of the HDRS17 as, despite generally recognized shortcomings of this scale, it is the gold standard for measuring depression. The manuscript is well written and I agree with all of the strengths outlined on lines 295-304. As a manuscript about the dimensionality of the HDRS17 I have little to criticize. However, it is a grave scientific weakness of the paper that it makes wide claims about valid measurement, which implies much more than dimensionality. As this issue is so pivotal in this paper, and as it is bound to mislead the readership of the journal on a very central issue to psychometrics, I will focus on this issue in the following.

Through the paper, one finds an apparent conflation between the summary scale score — i.e. they official, conventional and very simple way of deriving a single total score for the whole HDRS17 — and the highly elaborate, multidimensional person estimates derived from the structural model. We see a hint of this in the introduction, where the authors express concern that the total score of the HDRS17 may not measure the severity of a general depression factor. Yet they show no interest in testing a unidimensional measure. In the discussion (line 281), the authors state that "Such consistency in the factor structure of the HDRS17 [across two samples] would suggest that the HDRS17 is a valid measure of the general severity of depression in such settings". Here the authors again disregard the fact that the HDRS17 is conventionally scored based on the assumption of unidimensionality. However, the most manifest collision between the two confused concepts — the HDRS as a summary scale and the HDRS as it is modelled in ESEM — is found in the conclusions, line 319: "The total score of items on this measure [the HDRS17] are valid at each time point and over time". This is fundamentally wrong; a total score cannot be valid if it reflects more than one dimension. Also, the authors use a model without tau-equivalence, so their claim does not apply to the raw total score of the HDRS17, placing their validity claims far afield from the actual use of the HDRS17. Additionally, a valid total score builds on the assumptions of local independence and the absence of differential item functioning.

If the authors wish to say anything about the validity of the raw total score of the HDRS-17, they need to test (or at least evaluate) all the above assumptions. If, on the other hand, the authors wish to revolutionize the way the HDRS-17 is used, they need to explain to the reader how their model can be used for the purpose of valid measurement (i.e. entering every item score into a computer program which will then return three measures for each individual) and discuss benefits and limitations of this use over alternative uses (e.g. the conventional HDRS-17 summary score as well as shortened subscales with the potential for a statistically sufficient raw score, such as the HamiltonD6/melancholia subscale). Alternatively, if the authors are only concerned with the dimensionality of the scale, they should clearly state this as their aim and they should clarify to the reader that their results do not support most of the assumptions underlying the current use of the HDRS17.

Reviewer #2: 1) The authors identified individuals recruited in specialist mental health setting. It would be beneficial in understanding this population. Prevalence and severity of depression would help to identify why other measures were not chooses such as the BDI-II or the PHQ-9.

2) The authors mention inconsistencies of the HDRS17 factor structures even when compared to psychiatric interviews. The study’s methodology consisted of the Structured Clinical Interview (SCID) for DSM-IV. Analytical techniques using the SCID and the HDRS could have showed some consistency and validity of the data in terms of construct validity and reliability of the measure. There feels to be some missing processes known to be influential on the area of reported research. The study would benefit from analyses of reliability and validity across time stamps.

3) The authors evidenced fairly steady drops in the level of severity of depression. It would be beneficial to understand if treatment setting was moderating or mediating these values. Treatment with combination of CBT and psychopharmacological agent have shown the greatest reduction in depression severity. The authors study showed greater reduction of retardation-agitation factors score changes greater than worry-insomnia-factor. An elaboration of these results would be beneficial especially since factor loading of retardation-agitation are significant for depressed mood, suicidal thoughts, and agitations. It would have also been helpful if the results were comparable to other depression measures such as the PHQ-9 and BDI-II, and gain a better understanding of measurement precision and score comparability. The study in limited to the HDRS17 and it is difficult to really assess and control that the HDRS is capturing the whole spectrum of depression and it would be beneficial to apply some classical item response analysis.

4) Based on the above response and the novelty of the research study and inconsistencies related to the HDRS. The title of this study would be best fitted as a preliminary study of results.

6. PLOS authors have the option to publish the peer review history of their article (what does this mean?). If published, this will include your full peer review and any attached files.

Reviewer #1: Yes: Erik Vindbjerg

Reviewer #2: No

---

## [Author Response · Author response to Decision Letter 0]

18 Jun 2020

Response to PLOS ONE reviewers.

Reviewer comments on normal font and authors’ replies in bold and italics.

Reviewer #1: I welcome the initiative of the authors to explore psychometric properties of the HDRS17 as, despite generally recognized shortcomings of this scale, it is the gold standard for measuring depression. The manuscript is well written and I agree with all of the strengths outlined on lines 295-304. As a manuscript about the dimensionality of the HDRS17 I have little to criticize. However, it is a grave scientific weakness of the paper that it makes wide claims about valid measurement, which implies much more than dimensionality. As this issue is so pivotal in this paper, and as it is bound to mislead the readership of the journal on a very central issue to psychometrics, I will focus on this issue in the following.

Through the paper, one finds an apparent conflation between the summary scale score — i.e. they official, conventional and very simple way of deriving a single total score for the whole HDRS17 — and the highly elaborate, multidimensional person estimates derived from the structural model. We see a hint of this in the introduction, where the authors express concern that the total score of the HDRS17 may not measure the severity of a general depression factor. Yet they show no interest in testing a unidimensional measure. In the discussion (line 281), the authors state that "Such consistency in the factor structure of the HDRS17 [across two samples] would suggest that the HDRS17 is a valid measure of the general severity of depression in such settings". Here the authors again disregard the fact that the HDRS17 is conventionally scored based on the assumption of unidimensionality. However, the most manifest collision between the two confused concepts — the HDRS as a summary scale and the HDRS as it is modelled in ESEM — is found in the conclusions, line 319: "The total score of items on this measure [the HDRS17] are valid at each time point and over time". This is fundamentally wrong; a total score cannot be valid if it reflects more than one dimension. Also, the authors use a model without tau-equivalence, so their claim does not apply to the raw total score of the HDRS17, placing their validity claims far afield from the actual use of the HDRS17. Additionally, a valid total score builds on the assumptions of local independence and the absence of differential item functioning.

If the authors wish to say anything about the validity of the raw total score of the HDRS-17, they need to test (or at least evaluate) all the above assumptions. If, on the other hand, the authors wish to revolutionize the way the HDRS-17 is used, they need to explain to the reader how their model can be used for the purpose of valid measurement (i.e. entering every item score into a computer program which will then return three measures for each individual) and discuss benefits and limitations of this use over alternative uses (e.g. the conventional HDRS-17 summary score as well as shortened subscales with the potential for a statistically sufficient raw score, such as the HamiltonD6/melancholia subscale). Alternatively, if the authors are only concerned with the dimensionality of the scale, they should clearly state this as their aim and they should clarify to the reader that their results do not support most of the assumptions underlying the current use of the HDRS17.

We are grateful for the comments of reviewer 1, which stimulated considerable further thought. Following this we have substantially rewritten the introduction (paragraphs 2-5 and 7-9 of introduction) and discussion ( paragraphs 1-6, 6 and conclusion) sections, focusing on the factor structure of the scale, as a multidimensional measure. In keeping with this we have removed the previous references to ‘validity’ of the total score and have also removed the other specific lines identified. 

We have presented our results asa dimensional approach to the measurment of outcome in one UK cohort of patients with moderate-severe persistent major depression and made reference in the discussion to the importance of replication within other similar cohorts; and that results cannot be assumed to generalise to other less severe or persistent populations in other settings (such as primary care), referencing further work on this. 

Given the need for replication of the factor structure of the HDRS-17, we thought it important to also explicitly state that findings reported on change over 12 months are exploratory and have updated Table 5 to give further detail. We discuss some of the problems involved in operationalising this multidimensional model and the importance of replicating the bi-factor structure as a necessary prelude to this. With all of this in mind, as Reviewer 1 intimates, we are not in a position to revolutionize current use of the HDRS-17. We have instead made more modest calls for further work towards replication of our findings in similarly defined clinical groups (more severe persistent major depression). We have also discussed reasons why we cannot assume the observed factor structure to hold in other less severe, less persistent and likely less complex groups. 

Holding to the limitations above, we have also given some greater salience to comparisons with current 6-item scales (of Bech and Maier), including the different ways these scales were derived and some of the similarities in item content with the current model. Given the importance of replicating our current findings we have not though gone any further to suggest our bi-factor model is used in preference to these scales. 

As a result of these comments, the paper has been substantially re-written, particularly through the introduction and discussion sections and the line numbers have therefore changed but all quotes referenced by Reviewer 1 have been removed or substantially amended. Table 5 in the results section has also been re-drawn in keeping with the early stage of findings here and the more exploratory approach to what change over 12 months represents within this model. We think the manuscript is improved by these changes and are grateful to reviewer 1. 

Reviewer 2

Summary: The authors of this study explored the 17-item Hamilton Depression Rating Scale (HDRS-17) by examining the dimensionality of this measure with individuals diagnosed with moderately severe persistent major depressive disorder (PMDD). Results reported a bi-factor structure of two group factors (e.g., insomnia worry and retardation-agitation) that were stable over a 12-month period. The authors suggested that the HDRS-17 is a valid measure of depression for patients with moderately severe depression. 

Below I summarize my comments and concerns:

1) The authors identified individuals recruited in specialist mental health setting. It would be beneficial in understanding this population. Prevalence and severity of depression would help to identify why other measures were not chooses such as the BDI-II or the PHQ-9. 

We are grateful to Reviewer 2 and will answer this in parts:

1. All subjects were recruited from community mental health settings in the UK and with only half of these receiving additional collaborative care and the others continuing to receive general community mental health care. We have clarified this in the text ( method lines 121-123). 

2. Inclusion criteria from the initial study stated depression of at least moderate clinical severity, operationalised as HDRS-17 of 16 or over. We chose a mixture of clinician-rated outcome measures (CROMs; HDRS-17) and patient-rated outcome measures (PROMs; QIDS, PHQ-9 and BDI-1) considering that whilst we might expect a moderate-strong correlation, they could not necessarily be relied on to measure the same construct. The study team was aware that the different types of measure (CROMs and PROMs) might hold different perspectives and different points of reference (e.g. a clinical range vs personal experience) and so we considered them complimentary rather than replicative measures, both relevant to the study of Persistent Major Depression. The degree of correlation between CROMs and PROMs in this population is an interesting question and we have answered in more detail below, under point 3, comparing our findings with recently published material, though we have chosen not to include this data in the current manuscript in order to maintain a clearer focus on dimensionality of the HDRS17. We have though clarified some further detail from the initial study in the revised text ( methods lines 125-136).

2) The authors mention inconsistencies of the HDRS17 factor structures even when compared to psychiatric interviews. The study’s methodology consisted of the Structured Clinical Interview (SCID) for DSM-IV. Analytical techniques using the SCID and the HDRS could have showed some consistency and validity of the data in terms of construct validity and reliability of the measure. There feels to be some missing processes known to be influential on the area of reported research. The study would benefit from analyses of reliability and validity across time stamps. 

We have removed the reference to inconsistency with psychiatric interview, including any structured interview (e.g. SCID). We did not have strong enough data from the SCID to assess correlation but as shown in Table 1 (below), there was a moderate-strong correlation across the 3 time-points measured (Baseline, 6 and 12 months) between the HDRS-17 and three Patient Report Outcome Measures (PHQ-9, BDI, QIDS). The correlation was higher (within the moderate-strong range) for later timepoints, consistent with recent findings from Hershenberg et al. (2020) who showed greater correlation for patients under longer-term care (vs. patients at first contact). One might speculate on the reasons for greater agreement between clinician and patient outcome measures over time, such as developing trusting relationships, but the phenomenon seems consistent with other recent literature. Although we agree this is an important area, we would rather keep the current focus on factor structure and consider correlations in a later paper.

.

3) The authors evidenced fairly steady drops in the level of severity of depression. It would be beneficial to understand if treatment setting was moderating or mediating these values. Treatment with combination of CBT and psychopharmacological agent have shown the greatest reduction in depression severity. The authors study showed greater reduction of retardation-agitation factors score changes greater than worry-insomnia-factor. An elaboration of these results would be beneficial especially since factor loading of retardation-agitation are significant for depressed mood, suicidal thoughts, and agitations. It would have also been helpful if the results were comparable to other depression measures such as the PHQ-9 and BDI-II, and gain a better understanding of measurement precision and score comparability. The study in limited to the HDRS17 and it is difficult to really assess and control that the HDRS is capturing the whole spectrum of depression and it would be beneficial to apply some classical item response analysis.

We are grateful to Reviewer 2 for this comment and will answer in several parts to this, which are addressed separately below:

1. The specialist treatment (specialist CBT and psychopharmacology) was not dissociable from the treatment setting, indeed the integration of these approaches through ‘collaborative care’ was integral to the main study design. This intervention (collaborative care delivery of specialist CBT and psychopharmacology) was associated with measurable changes during treatment (not present at baseline) and we therefore consider treatment as a mediator (rather than moderator) of outcome. Although the suggestion of a mediator analysis is interesting, it is outside the scope of the current study, on HDRS-17 dimensionality and we are reluctant to change the focus of the current paper. However, we thank reviewer 2 for stimulating thought on this and will consider a future path analysis, assessing correlated change of transdiagnostic measures with the separate HDRS factors shown here. 

2. We agree the greater reduction in the retardation-agitation factor is of relevance clinically and have added to both the discussion (lines 250-275) and conclusion sections to reflect this. We have given further detail on this change in factor score over 12 months (Table 5) but have discussed it as an exploratory analysis; with need to replicate the basic bi-factor structure and then work towards operationalizing findings.

3. We have performed a correlation analysis, presented below in Table 1, which shows significant and moderate-strong correlation between the observer rated HDRS-17 and all patient completed outcome measures used in this study (PHQ-9, BDI-1 and QIDS, at each time point). Correlation shown in this analysis was similar to that reported in another recent publication in a Treatment Resistant Depression population (Hershenberger et al, 2020), particularly for the closest comparison (at baseline comparison, before any clinical improvement). 

Table 1

 HDRS-17 Current Study HDRS-17

 Baseline 6 months 12 months Reported*

BDI-1 0.56 0.75 0.78 0.65

QIDS-SR 0.55 0.69 0.76 0.57

PHQ-9 0.49 0.68 0.79 Not Available

*From Hershenberg et al, Journal of Affective Disorders, April 2020

4. Our study aimed to assess the dimensionality of the HDRS-17 measure with preliminary evidence of a bi-factor structure fitting the data well. We did not hypothesise that the HDRS-17 measures a single concept but conducted an exploratory analysis that identified a multidimensional construct, with specific factor scores indicating the severity of relevant aspects of depression. In the initial RCT, we did not hypothesise that the HDRS-17 would capture the whole spectrum of depression and included a number of scales therefore (PHQ-9, BDI and QIDS-SR) as complimentary, rather than strictly replicative measures that might give a fuller understanding, both from different perspectives (patient vs clinician) and slightly different concepts of ‘depression’ (for example including appetite gain, as well as loss in QIDS-SR). Although we accept the point made here, again we would rather keep the focus of the current paper on the important issue of the dimensionality of the HDRS-17. 

4) Based on the above response and the novelty of the research study and inconsistencies related to the HDRS. The title of this study would be best fitted as a preliminary study of results. 

Thank you for this recommendation, which we have accepted in full.

The full title now reads “The bi-factor structure of the 17-item Hamilton Depression Rating Scale in moderate to severe persistent major depression; dimensional measurement of outcome”

We think this better represents the early findings reported here that need replication and we’re grateful to Reviewer 2 for the stimulating comments, which have hopefully improved the manuscript.

---

## [Decision Letter · Decision Letter 1]

15 Jul 2020

PONE-D-20-01015R1

The bi-factor structure of the 17-item Hamilton Depression Rating Scale in persistent major depression; dimensional measurement of outcome.

PLOS ONE

Dear Dr. Morriss,

Thank you for submitting your manuscript to PLOS ONE. After careful consideration, we feel that it has merit but does not fully meet PLOS ONE’s publication criteria as it currently stands. Therefore, we invite you to submit a revised version of the manuscript that addresses the points raised during the review process.

We look forward to receiving your revised manuscript.

Kind regards,

Thach Duc Tran, M.Sc., Ph.D.

Academic Editor

PLOS ONE

Reviewers' comments:

Reviewer's Responses to Questions

**Comments to the Author**

1. If the authors have adequately addressed your comments raised in a previous round of review and you feel that this manuscript is now acceptable for publication, you may indicate that here to bypass the “Comments to the Author” section, enter your conflict of interest statement in the “Confidential to Editor” section, and submit your "Accept" recommendation.

Reviewer #1: (No Response)

Reviewer #2: All comments have been addressed

2. Is the manuscript technically sound, and do the data support the conclusions?

Reviewer #1: Partly

Reviewer #2: Yes

3. Has the statistical analysis been performed appropriately and rigorously? 

Reviewer #1: N/A

Reviewer #2: Yes

4. Have the authors made all data underlying the findings in their manuscript fully available?

Reviewer #1: No

Reviewer #2: Yes

5. Is the manuscript presented in an intelligible fashion and written in standard English?

Reviewer #1: Yes

Reviewer #2: Yes

6. Review Comments to the Author

Reviewer #1: The authors have used the feedback to substantially rewrite parts of the introduction, discussion, and conclusion section, "focusing on the factor structure of the scale, as a multidimensional measure". In their response to my previous comments, they explain how they have now toned down the implications of their results, e.g. that they now refrain from referring to the validity of the total score, and they explicate that their results need replication before they can be generalized. In fact, the authors now say nothing about validity, and they focus on how the results should first be replicated before we look at the challenges of operationalising — and presumably validating — the model for measurement. Here, it would have been helpful if the authors had chosen more explicitly amount the three routes outlined in my previous review: (a) to deal with the validity of a raw total score, (b) to revolutionize the way we use the HDRS-17, and (c) to deal exclusively with the dimensionality. While the authors refrain from a, they still stray between path b and c.

The authors include a comparisson of their model with the six-item subscales defined by Bech and Maier, respectively. This is particularly welcome, as the Melancholia subscale has received support for construct validity, i.e. it appears that it can be utilized for measurement. This would be a good place to establish, how the authors consider the potential validity of their bi-factor model of the HDRS. The authors promote their model, contingent on replication, by writing that "This multidimensional reporting might provide a more accurate guide to future research and treatment in moderate-severe PMDD, including outcome monitoring". As such, they call for a fundamental change of the way the HDRS is used (what I refer to as "path b" above). This raises a host of questions about how the scale could be used and how it would need to be validated for such use. Strictly speaking, acknowledging the multidimensionality of the HDRS may allow us to retrieve more information from the responses, but when dealing with an essentially psychometric manuscript, we would not expect the authors to dismiss or postpone the issue of validity. As an example, when the authors allow negative loadings on two factors, we need to clarify whether person estimates for these traits can be used for outcome measurement.

A second point to make about the Melancholia subscale, is that it resembles the general depression factor of this study. The authors relate it to the retardating/agitation factor, while, in fact, five out the the six items of the Melancholia scale display their highest loading on the general depression factor. The authors need to address this discrepancy. It also puts them in a better position to compare the utility of the Melancholia subscale and their General depression factor used as a scale for outcome measurement.

As a separate, technical issue, I would like to ask the authors to clarify whether they included residual correlation parameters for the three imsomnia related items. Given the well-proven instability of the HDRS, and the authors' call for replication, it is important to have full transparency of the detection and modelling of any residual correlations, theorized ones as well as empirically discovered ones.

I will reserve a final comment for inspiration, which may guide the authors in their reflection of the issues pointed out above. Here, I will try and explicate my own understanding of the implications of suggesting the bifactor model for clinical use. First, the factor structure of the HDRS has prooven highly unstable, and we need to put a lot of faith into the bifactor model to believe this is now going to change. But we cannot deny that it is possible, so I've raised no issue with this. Second, if we want to establish construct validity, we would likely need to do an elaborate analysis based on multidimensional item response theory, which would allow for testing of differential item functioning. Again, we cannot deny that this is possible despite the highly elaborate model, so I have raised no issue with this. Then, for clinical utility, we would need to enter each response into software that would return a person estimate for each factor based on the psychometric model. I wonder if the authors are aware of this. Based on the loadings in Table 3, we would expect to see the highest reliability on the General depression estimate, and as this factor has no negative loadings, it can potentially be used for outcome measurement. Arguements can be made for and against this estimate, relative to the simpler and well supported estimate of the Melancholica subscale, or even the 10-item Bech and Raphaelsen scale, which cover additional ICD-10 items of Major Depression (including suicidal ideation). The remaining two subscales cannot be used for outcome measurement, but as the authors suggest, may be used for outcome preduction for different types of intervention. Reliability may be limited, so for patients within a potentially wide confidence interval no clear guidance will be given. Given this fragmentation of the scale into an outcome scale and two predictor scales, we may ask ourselves why we are so keen to stick with the particular pool of items included in the HDRS-17.

Reviewer #2: The authors have addressed concerns noted in first reviewer comments. I would have liked to see more in-depth methodology consisting of analytical techniques using the SCID, though the authors commented limited data. It would have been refreshing to see how associated or related the HDRS-17 would have been in addressing the criterion of persistent depression with other measures such as the PHQ-9, BDI-II, as this in my opinion would have added more clarity and depth regarding the dimensionality of the HDRS-17. However, authors were able to address my comments and concerns and I believe this is a publishable piece of work. I also appreciated a title change.

7. PLOS authors have the option to publish the peer review history of their article (what does this mean?). If published, this will include your full peer review and any attached files.

Reviewer #1: **Yes: **Erik Vindbjerg

Reviewer #2: No

---

## [Author Response · Author response to Decision Letter 1]

25 Aug 2020

Response to Reviewer 1

1. We were asked to deal with the potential validity of the bi-factor model in the context of the Melancholia (Bech, Maier) subscale - that this should not be postponed and the specific example of negative loadings needs to be addressed, including whether person estimates for these traits can be used for outcome measurement?

We re-considered the implications of the HDRS17 bi-factor structure shown in our analysis. We agree that this is not appropriate for use as an outcome measure in the moderate-severe PMDD group studied here, given that it is not a unidimensional measure and that domain factors include negatively loading items. In light of this and previous evidence on the HDRS6 (Bech), we conducted a further post-hoc analysis to assess the dimensionality of the HDRS6 subscale in this group. This analysis is given below and included as a post-hoc analysis in our updated results section. It shows a significant positive loading of all 6 items (0.396 – 0.544, p < 0.01) for this moderate-severe PMDD group, with time invariance. Considering this fuller evidence within the wider context advocated by Reviewer 1, we have revised our discussion, proposing the HDRS6 as an outcome measure in moderate-severe PMDD. The HDRS17 may retain a clearer role in psychiatric assessment where its more complex structure identifies types of PMDD, acting as a potential predictor of effective treatment strategies for settled states of depression (e.g. for agitated vs. retarded depression). 

2. Address why we have related the melancholia subscale to the ‘retardation-agitation’ domain rather than the general factor, when 5 out of 6 items on the melancholia subscale (Bech) display highest loadings on the general factor?

The HDRS-6 comprises depressed mood, anhedonia (work and interests), guilt, fatigue, psychological anxiety and psychomotor retardation (Bech 1981); and as Reviewer 1 points out, 5 of these 6 items showed their highest loading on the general factor, rather than either of the domain factors. Prompted by this, we ran a further analysis investigating the structure of the HDRS6 in our PMDD cohort, specifically to investigate whether this showed a unidimensional structure and time-invariance. For this reason (and because it is not plausible to do exploratory analysis testing model with 1 to 3 factors on a 6-item scale), we focused on testing the unidimensional factor structure only, i.e. we used a one factor model with results given below:

Item HDRS6

Depressed Mood .544*

Work and Interests .526*

General Somatic (Tiredness) .474*

Psychic Anxiety .417*

Gulit Feelings .396*

Psychomotor retardation .407*

* all loading estimates are statistically significant at p<0.01

The HDRS6 also showed stability over time:

Table Bech scale Fitting indices of ME/I across measurement time

Model �2(df),p= RMSEA CFI NNFI ΔCFI Δ�2(df),p=

Configural 193.266(120),0.000 .057 .934 .913 

Scalar a 459.218(165),0.000 .098 .736 .755 259.790(45), p=0.000

Scalar b 229.625(146),0.000 .055 .925 .921 -.009 44.883(26), p=0.012

Note: scalar b model freed 24 of 55 (43%)threshold parameters

The unidimensionality, positive item loading and stability of the HDRS6 indicate it may be used as an outcome measure in moderate-severe PMDD. Additionally it is more parsimonious than the HDRS17, therefore quicker to administer and much more likely to be of clinical use as an outcome measure. We’re grateful for Reviewer 1 pointing us in this direction and have changed our discussion to indicate the potential use of the HDRS17 within assessment, with its more complex structure helping to predict effective treatment interventions; with evidence-based item-sets such as the HDRS6 used for outcome measurement. This approach seems consistent with Hamilton’s (1960) own assessment of the multidimensionality of the HDRS17 and its use in measuring settled states, rather than treatment change.

3. Did we include residual correlation parameters for the three insomnia related items? 

No, as correlating residuals is generally not allowed when running SEM models and would only have been justifiable for repeated items or MTMM items when Items were analysed as a different set of factors.

4. Given the instability of the HDRS factor structure, will the bi-factor model change this? 

No, we agree this won’t save the HDRS17 as a measure of outcome change. The bi-factor structure may remain useful to clinicians and researchers assessing settled states of depression (rather than assessing change in state) and we have added to the discussion of this.

5. Only the General factor can be used as an outcome measurement, with the subscales (containing neg loadings) only as predictor factors (e.g. for different types of interventions being more helpful) – given this why even stick with the particular set of questions in the HDRS-17?

We have carefully considered this. Given our results on the factor structure of the HDRS-17 in this well-defined clinical population (moderate-severe PMDD), we consider a more evidence-based position would be to recommend use item sets such as the HDRS6 in outcome measurement, with the HDRS17 having a more restricted use, within assessment of settled states (where the more complex structure may be an advantage in developing treatment strategies). We are very grateful for the helpful comments of Reviewer 1 in reconceptualising these results.

---

## [Decision Letter · Decision Letter 2]

8 Sep 2020

PONE-D-20-01015R2

The bi-factor structure of the 17-item Hamilton Depression Rating Scale in persistent major depression; dimensional measurement of outcome.

PLOS ONE

Dear Dr. Morriss,

Thank you for submitting your manuscript to PLOS ONE. After careful consideration, we feel that it has merit but does not fully meet PLOS ONE’s publication criteria as it currently stands. Therefore, we invite you to submit a revised version of the manuscript that addresses the points raised during the review process.

We look forward to receiving your revised manuscript.

Kind regards,

Thach Duc Tran, M.Sc., Ph.D.

Academic Editor

PLOS ONE

Reviewers' comments:

Reviewer's Responses to Questions

**Comments to the Author**

1. If the authors have adequately addressed your comments raised in a previous round of review and you feel that this manuscript is now acceptable for publication, you may indicate that here to bypass the “Comments to the Author” section, enter your conflict of interest statement in the “Confidential to Editor” section, and submit your "Accept" recommendation.

Reviewer #1: All comments have been addressed

2. Is the manuscript technically sound, and do the data support the conclusions?

Reviewer #1: Yes

3. Has the statistical analysis been performed appropriately and rigorously? 

Reviewer #1: Yes

4. Have the authors made all data underlying the findings in their manuscript fully available?

Reviewer #1: No

5. Is the manuscript presented in an intelligible fashion and written in standard English?

Reviewer #1: Yes

6. Review Comments to the Author

Reviewer #1: I commend the authors on a well executed revision to their manuscript. My only additional request, is that the authors add a brief clarification in their manuscript, that they did not find any substantial residual correlations in the bi-factor model, as this would violate a basic assumption of the model. Also, optionally, I may recommend the authors to state that their results for the HDRS-6 do not establish if the total score features statistical sufficiency, and as such individual items may need to be weighed differently (e.g. higher weight to depressed mood, lower weight to psychomotor retardation, etc.). This is simply to underline that a scale is not simply valid or not, nor simply valid with a perticular population or not, but that validity also depends on the use case, e.g. whether a raw total score is used or if individual item responses are loaded differently in the scoring, in accordance with their loading in the CFA.

7. PLOS authors have the option to publish the peer review history of their article (what does this mean?). If published, this will include your full peer review and any attached files.

Reviewer #1: **Yes: **Erik Vindbjerg

---

## [Author Response · Author response to Decision Letter 2]

12 Oct 2020

Response to reviewers.

We are grateful for the further comments, which are addressed specifically below. Additionally, we have included the data requested. 

Review Comments to the Author

6. “I commend the authors on a well executed revision to their manuscript. My only additional request, is that the authors add a brief clarification in their manuscript, that they did not find any substantial residual correlations in the bi-factor model, as this would violate a basic assumption of the model”

We thank Reviewer 1 for the further comments. It is not a recommended convention to correlate error terms when preforming CFA and SEM [1-3] and we therefore set the correlation between item residuals as 0, as per Mplus default setting for ESEM. For complete clarity we have added to lines 174/5 in the text: All ESEM models were performed using software Mplus 8 and in keeping with standard practice correlation between item residuals was set as 0. 

“Also, optionally, I may recommend the authors to state that their results for the HDRS-6 do not establish if the total score features statistical sufficiency” 

We discussed this option within the team and whilst tempting, we concluded that additional comment on a post-hoc analysis would take the discussion too far from the central findings. However, in writing this we agree that future work with a main focus on statistical sufficiency (or indeed alternative item inclusion) is warranted.

1. McDonald R, Ho M-H. Principles and practice in reporting Structural Equation Analyses. Psychological Methods. 2002;7:64-82. doi: 10.1037/1082-989X.7.1.64.

2. Schreiber JB, Nora A, Stage FK, Barlow EA, King J. Reporting Structural Equation Modeling and Confirmatory Factor Analysis results: A review. The Journal of Educational Research. 2006;99(6):323-38. doi: 10.3200/JOER.99.6.323-338.

3. Hermida R. The problem of allowing correlated errors in Structural Equation Modeling: Concerns and considerations. Computational Methods in Social Sciences. 2015;3:1-17.

---

## [Editor Report · Decision Letter 3]

14 Oct 2020

The bi-factor structure of the 17-item Hamilton Depression Rating Scale in persistent major depression; dimensional measurement of outcome.

PONE-D-20-01015R3

Dear Dr. Morriss,

We’re pleased to inform you that your manuscript has been judged scientifically suitable for publication and will be formally accepted for publication once it meets all outstanding technical requirements.

Kind regards,

Thach Duc Tran, M.Sc., Ph.D.

Academic Editor

PLOS ONE
---

## [Editor Report · Acceptance letter]

16 Oct 2020

PONE-D-20-01015R3 

The bi-factor structure of the 17-item Hamilton Depression Rating Scale in persistent major depression; dimensional measurement of outcome. 

Dear Dr. Morriss:

I'm pleased to inform you that your manuscript has been deemed suitable for publication in PLOS ONE. Congratulations! Your manuscript is now with our production department. 

Kind regards, 

on behalf of

Dr. Thach Duc Tran 

Academic Editor

PLOS ONE